# Role of Vitamin C in Targeting Cancer Stem Cells and Cellular Plasticity

**DOI:** 10.3390/cancers15235657

**Published:** 2023-11-30

**Authors:** Yool Lee

**Affiliations:** 1Department of Translational Medicine and Physiology, Elson S. Floyd College of Medicine, Washington State University, Spokane, WA 99202, USA; yool.lee@wsu.edu; 2Steve Gleason Institute for Neuroscience, Washington State University, Spokane, WA 99202, USA

**Keywords:** vitamin C, cancer stem cells, epithelial–mesenchymal transition

## Abstract

**Simple Summary:**

This review is centered on the potential therapeutic utility of vitamin C (VC) in the context of dynamic cancer evolution. Physiologically, VC has dose-dependent roles as both an antioxidant and a pro-oxidant and has been associated with various health benefits, including potential applications in cancer management. Its intriguing ability to selectively target cancer cells has sparked recent research exploring its potential use against cancer stem cells (CSCs), a crucial player in both tumorigenesis and metastasis. CSCs play a pivotal role in tumor progression, metastasis, and resistance to conventional drug treatments. This review seeks to elucidate the underlying mechanisms by which VC effectively targets and reduces the activity of CSCs, thereby providing important insights and rationale for the improvement of anti-cancer therapeutic strategies.

**Abstract:**

Vitamin C (VC) is an essential nutrient that is vital for maintaining cellular physiology. Interestingly, it functions as either an antioxidant or a pro-oxidant, depending on the concentration used. At high-doses, VC selectively targets various cancer cell types through its pro-oxidant action, while at low-doses, VC enhances anti-tumor immunity by acting as an antioxidant. This versatility makes VC a promising anti-tumor agent for both standalone and combination therapies. Tumors consist of diverse cancer cell subtypes with distinct phenotypic and functional characteristics. In particular, cancer stem cells (CSCs), which are self-renewing multi-potent cells, are responsible for tumor recurrence, metastasis, chemoresistance, and heightened mortality. CSCs are often associated with the epithelial–mesenchymal transition (EMT), which confers increased motility and invasive capabilities that are characteristic of malignant and drug-resistant cells. Thus, eradicating CSC populations is crucial and has led to extensive efforts aimed at identifying medicines that can target them. Recent studies suggest that VC can selectively target CSCs via epigenetic and metabolic pathways in various cancers. Here, we highlight recent progress that has been made in understanding how VC effectively targets CSC evolution, providing a rationale for the use of VC either alone or in combination with other treatments to improve outcomes.

## 1. Introduction

Vitamin C (L-ascorbic acid, ascorbate, VC) is an essential nutrient for the normal maintenance of cellular functions, such as neural pathways, molecule biosynthesis (e.g., collagen, norepinephrine), immune signaling, chromatin remodeling, and cell division [1,2,3,4]. At physiological concentrations (40–80 µM in human plasma), VC acts as an antioxidant by serving as an electron donor and effectively scavenges reactive oxygen species (ROS). In contrast, at high doses (10–20 mM), VC acts as a pro-oxidant that induces oxidative stress and suppresses tumor growth, without notable damage to normal cells and tissues [5,6]. Recently, multiple studies have uncovered that VC has multifaceted anti-tumor effects [7]. For example, it acts as a cofactor for enzymes that regulate gene expression and suppresses oncogenes while reactivating tumor suppressor genes [8,9,10]. It has also been reported that VC induces various forms of canonical cell death and may be involved in non-canonical mechanisms relating to energy crises resulting from ATP depletion [11]. Aligned with its pleiotropic effects as a cancer-specific, pro-oxidative cytotoxic agent, anti-cancer epigenetic regulator, and immune modulator, high-dose VC has been proposed as a potent adjuvant treatment for cancer, acting synergistically with numerous standard (chemo-) therapies and alleviating the toxic side effects of chemotherapy (refer to for comprehensive reviews [12,13] of this). Furthermore, emerging animal model studies suggest that VC enhances anti-tumor effects when used in combination with dietary intervention (intermittent fasting, IF) [14] or cancer immunotherapy that facilitates anti-tumor immune environments [15,16], suggesting that VC mediates beneficial anti-cancer effects by targeting both tumor-intrinsic and -extrinsic pathways [17].

One of the most critical issues in cancer biology, as well as cancer diagnosis and treatment, is cancer cell plasticity, the adaptive and reversible capacity of diverse cancer cell populations to shift between cancer stem cell (CSC) and non-CSC/differentiated cell states in response to the tumor microenvironment [18]. At the top of the heterogeneous tumor hierarchy, the CSC, a self-renewing and multi-potent cancer cell type, is responsible for tumor recurrence, metastasis, chemoresistance, and mortality [18]. Phenotypically, CSCs are associated with the epithelial–mesenchymal transition (EMT), which confers cancer cells with increased motility and invasion ability that is characteristic of malignant and drug-resistant cells [19]. A growing number of studies suggest that CSC heterogeneity and plasticity are influenced not just by genetic factors but also by non-genetic factors, including epigenetic pathways [20], metabolic processes [21], and tumor-microimmune environments [22,23,24]. CSCs develop resistance to traditional chemotherapy due to their adaptable phenotype, which enables them to withstand therapy by overexpressing anti-apoptotic factors, defending against oxidative stress, and effectively repairing DNA damage [25]. Unfortunately, the vast majority of anti-cancer drugs have been designed to target rapidly dividing non-CSCs, rather than dormant CSCs, frequently leading to tumor relapse and treatment ineffectiveness [26,27]. Furthermore, many chemotherapy drugs trigger diverse mechanisms of plasticity in cancer cells, including EMT, autophagy, and metabolic reprogramming, contributing to the evolution of therapy-resistant tumors [28]. While extensive pharmacological efforts have been made to specifically target CSCs, with the aim of eradicating this malignant and drug-resistant cell population [29,30,31], many synthetic drugs have toxic effects on normal tissues and their use can be accompanied by several detrimental side effects on physiology and behavior [32]. This underscores the need for safer and more targeted therapeutic approaches that can effectively eliminate CSCs without causing significant harm to normal tissues, thus minimizing adverse effects. In this context, there is a growing recognition that less toxic natural products, possessing anti-cancer stem cell (CSC) activities, such as flavonoids, FDA-approved drugs derived from natural sources, and nutritional herbs commonly employed in traditional Chinese medicine, hold promise as potential alternatives for addressing therapy-resistant cancers [33,34].

In recent years, an increasing body of research has indicated that VC has a preferential ability to target CSC populations by modulating epigenetic and metabolic pathways in various cancer types, including leukemia [35,36], liver cancer [37,38], and breast cancer [39,40]. Moreover, a more recent study has shown that pharmacological VC enhances the effectiveness of combination nanomedicines and reduces cancer cell stemness, thus preventing post-surgery recurrence and systemic metastasis [41]. In this review, we will focus on recent advancements in our understanding of how VC, used as a standalone treatment or in combination with other anti-cancer strategies, can efficiently target CSC evolution, with the aim of offering important insights and a rationale for utilizing VC to improve cancer therapy and prognosis.

## 2. Physiological and Anti-Tumor Activities of Vitamin C

Physiologically, VC exists largely in its reduced (ascorbic acid [AA]) or oxidized (dehydroascorbic acid [DHA]) forms, which, depending on its redox state, involves the loss or gain of two electrons [7] (Figure 1A).

Notably, VC exhibits several drug-like properties that make it a promising therapeutic agent. VC has a low molecular weight (<500 Da, 176.12 g/mol), is water-soluble, and has a high oral bioavailability. The bioavailability of VC in foods is generally considered equivalent to the purified form within the recommended nutritional range of 15–200 mg [45]. Yet, this bioavailability diminishes by over 50% with higher amounts, such as doses exceeding 1000 mg. Since VC was first chemically synthesized in 1933, the bioavailability of synthetic and natural VC has been a subject of extensive research [46]. Animal studies indicate varying bioavailability between synthetic and natural VC, depending on the study design and animal model. In contrast, human studies have consistently shown no significant differences in their bioavailability [47]. While synthetic and natural VC share the same molecular and physicochemical properties, it has been reported that fruits and vegetables offer a wealth of micronutrients, dietary fiber, and phytochemicals that can modulate the absorption and utilization of VC [47,48]. Furthermore, VC is highly susceptible to oxidation and degradation, particularly in biological fluids such as plasma and blood [47,48]. The stability of VC in these fluids is influenced by several factors that can occur during processing (e.g., heat and light, pH, metal ions) and storage (e.g., temperature, oxygen exposure). Particularly, in vivo VC levels are determined by a balance between uptake, metabolism, and excretion (refer to [49] for a comprehensive review of these processes). Notably, the conversion of ascorbic acid to DHA in foods or the gastrointestinal tract can diminish the bioactivity of VC [49]. To enhance the chemical stability and bioavailability of VC, various chemically synthesized ascorbic analogs, such as ascorbate 2-sulfate, ascorbate 2-monophosphate, and ascorbate 2-triphosphate, have been developed [45]. Additionally, encapsulating VC in specific nanoparticles has been shown to improve stability during storage and delivery [45].

The transport of the reduced form of VC (AA) occurs through specialized transporters known as sodium-dependent vitamin C transporters (SVCT) 1 and 2, which are conserved across mammalian species, including humans [50,51,52,53] (Figure 1B). Notably, it has been reported that SVCT2 is a key protein for VC uptake in both normal [54,55] and cancer cells [37,56]. Moreover, VC has been shown to inhibit breast cancer cell growth [56] or preferentially kill CSC populations in live cancer in an SVCT2-dependent manner [37]. In contrast, transport of the oxidized form of VC (DHA) into cells is primarily facilitated by glucose transporters known as GLUTs (GLUTs 1–4 and 8) [57,58,59,60]. According to the prevailing model, DHA is the most active anti-cancer form of VC in tumors, as it generates cytotoxic reactive oxygen species (ROS) upon its intracellular conversion to AA following its entry into cells [61,62]. However, recent studies involving direct treatment of various cancer cell lines, including human breast cancer and neuroblastoma cells, with DHA have consistently shown that DHA has minimal or no significant impact on cell death [42,43,63]. This suggests the possibility that the cytotoxic responses to DHA can vary depending on the specific cancer cell type or experimental conditions. Taken together, these results underscore the need for further studies in a wide range of cancer cell types to explore how the distinct redox forms of VC contribute to VC-induced cell death.

Numerous preclinical investigations of various human cancer models have indicated that the extracellular generation of hydrogen peroxide (H_2_O_2_) is a pivotal factor in the anti-cancer efficacy of high-dose VC [13,63]. AA readily oxidizes to DHA through a two-electron oxidation process in the presence of catalytic metals, like copper (Cu^+^/Cu^2+^) and iron (Fe^3+^/Fe^2+^), leading to elevated H_2_O_2_ concentrations in the extracellular space of tumors [7,64,65]. The H_2_O_2_ can subsequently permeate cells by utilizing peroxiporins within the plasma membrane to exert its influence on redox-dependent signaling and metabolic pathways pertinent to the viability of cancer cells, including the pathways regulating processes such as cell-cycle arrest, DNA damage, and apoptosis [66] (Figure 1B). However, the role of iron in the anti-cancer action of VC has recently been debated, with varying findings in different in vitro studies. Some investigations have observed that reducing or depleting intracellular iron levels enhances the growth inhibition and apoptosis induced by VC in neuroblastoma and K562 leukemic cells [67,68], while others have reported that extracellular iron diminishes the anti-cancer effects of VC in PC-3 and LNCaP prostate cancer cell lines [69]. More recently, it has been documented that exogenous iron impairs the anti-cancer effects of VC in specific cancer cell lines, both in vitro and in vivo [70]. These findings suggest that the impact of iron on VC-induced cytotoxicity may vary depending on the cell type or experimental method employed, such as inhibiting intracellular iron using iron chelators or exogenous iron treatment, necessitating further exploration in other cancer types.

Notably, the mechanisms of cell death underlying the anti-cancer effects of VC have undergone extensive investigation. Previous research has suggested that pharmacological VC can trigger various forms of cell death, including apoptosis, necroptosis, and autophagy, with the specific outcome contingent upon the concentration and cell type employed in the experiment [43]. Earlier studies have also indicated that VC-induced cytotoxicity is primarily mediated through caspase-dependent apoptosis or necrosis, based on assessments of changes in the protein levels of key cell death effectors, such as caspases, BAX, BID, and receptor-interacting protein kinase (RIPK1), in response to VC exposure [71,72,73]. However, emerging evidence indicates that classical inhibitors of apoptosis or necrosis, such as the pan-caspase inhibitor Z-VAD-FMK and the RIP1-targeted necroptosis inhibitor Nec-1, do not prevent the cell death induced by pharmacological VC, suggesting that there are non-canonical cell death mechanisms at play [74,75]. Recently, non-apoptotic forms of cell death, such as ferroptosis, parthanatos, and pyroptosis, have garnered attention as promising targets for cancer therapy with natural or synthetic compounds that induce ROS [76]. These findings suggest that the cytotoxic effects induced by high-dose VC may entail multiple cell death pathways operating synergistically, rather than a single pathway.

In addition to cell death mechanisms, metabolic crises are a recurring phenomenon in cancer cell death triggered by pharmacological VC treatment [18]. Consistent with the Warburg hypothesis, which indicates cancer cells’ preference for glycolysis over oxidative phosphorylation for energy production, ATP depletion and cell demise in response to VC are primarily attributed to the hindered glycolysis caused by the VC-induced H_2_O_2_-mediated inhibition of glyceraldehyde 3-phosphate dehydrogenase (GAPDH) activity [43,62]. In this context, recent studies have proposed a model suggesting that VC-induced H_2_O_2_ inflicts DNA damage, consequently promoting poly (ADP-ribose) polymerase (PARP) activation, which, in turn, consumes NAD and depletes ATP through the reduction of GAPDH activity and glycolysis [43,61,77,78]. However, subsequent investigations have reported that treatment with the PARP inhibitor Olaparib maintains NAD^+^ and ATP levels but results in increased DNA double-strand breaks and does not prevent ascorbate-induced cell death [43]. This suggests that the PARP-associated DNA damage response may not be the exclusive cause of this cytotoxicity, implying that supplementary mechanisms contribute to the NAD^+^- and ATP depletion-dependent cytotoxicity of VC treatment [43] (Figure 1B).

In addition to its pro-oxidant properties, several studies have uncovered additional VC-mediated anti-tumor mechanisms involving epigenetic and post-translational pathways (Figure 1C). VC has been found to serve as a key cofactor that catalyzes the activity of various iron-containing dioxygenase enzymes, such as ferrous iron Fe^2+^ and α-ketoglutarate (αKG)-dependent dioxygenases (Fe^2+^/α-KGDDs), that play diverse roles in many biological processes, including the regulation of metabolic adaptations to hypoxia, the epigenetic regulation of gene transcription, and the reprogramming of cellular metabolism [7]. These enzymes include hypoxia-inducible factor (HIF) hydroxylases (e.g., prolyl-hydroxylase domain-containing proteins (PHDs 1–3) and factor inhibiting HIF (FIH)), as well as DNA demethylases (e.g., TET1–3) [8,9,10]. The activities of these enzymes contribute to the suppression of oncogenes and the re-expression of tumor suppressor genes, resulting in post-translational and/or epigenetic anti-tumor effects in both hematological and solid tumors [36,79,80]. Indeed, a growing number of studies report the involvement of these enzymes in VC-induced tumor suppression across multiple cancer types, including leukemia, melanoma, and renal cell carcinoma [4,79,80,81,82].

Harnessing its multifaceted effects in cancer, high-dose VC has emerged as a promising therapeutic strategy, either as a standalone treatment or in combination with various standard (chemo-) therapies, potentially alleviating the toxic side effects associated with chemotherapy [12,13,83]. Notably, even intravenous administration of very high doses of VC, ranging from 1 to 200 g and administered repeatedly, was reported to be well tolerated in the majority of patients [84]. However, caution has been noted regarding the administration of high doses, as they may lead to overt side effects in certain susceptible patients, such as the formation of oxalate renal stones [85]. Moreover, it is important to note that some patients may experience side effects, including diarrhea, nausea, abdominal cramps, and other gastrointestinal issues [85]. In light of these considerations, national clinical trials (NCT) have been actively investigating the effects of vitamin C as a standalone treatment or in combined therapies across various cancers, including EGFR mutant non-small cell lung cancer (NSCLC) (NCT04033107), recurrent high-grade glioma (NCT01891747), metastatic colorectal cancer (NCT04516681, NCT02969681), KRAS and BRAF mutant colon cancer (NCT04035096), hepatocellular carcinoma, pancreatic cancer, gastric cancer, colorectal cancer (NCT04033107), and acute myeloid leukemia (AML) (NCT02877277) (refer to a recent comprehensive review [17] for further details on this).

## 3. Cancer Stem Cell Phenotypes and Plasticity

Stem cells are a specialized group of cells that have the capacity to differentiate into various cell types within the body; thus, they play crucial roles in tissue development, differentiation, and the maintenance of overall physiological balance [86]. With the increasing characterization of stem cell-specific markers, as well as the development of lineage tracing (e.g., barcode technology, single-cell RNA-sequencing) and three-dimensional (3D) organoid technologies in the field of stem cell research, it has become increasingly evident that, akin to normal tissues, cancer cells within tumors are not uniform but are instead diverse cell populations with distinct cellular lineages and properties [26]. Furthermore, diverse sophisticated assays have been devised for the isolation of CSC populations. These include fluorescence-assisted cell sorting of Rhodamine-123 positive side populations, detection of cell-membrane-specific antibody positivity, drug-efflux-based assays, identification of aldehyde dehydrogenase-1 (ALDH-1)-positive cells, and isolation of drug-resistant phenotypes [87]. Positive selection of resistant phenotypes in the presence of cytotoxic drug concentrations has been effectively utilized for isolating drug-resistant stem cells [34,88]. These cells exhibit characteristics such as stem-cell-selective tumor spheroid formation, cell surface molecules, and nuclear transcription factors. The status of stem cell markers is quantified through tumor spheroid formation and the expression of select molecules, including clusters of differentiation clusters of differentiation CD44 and CD133, nuclear transcription factors octamer-binding transcription factor-4 (OCT-4), sex determining region box Y-2 (Sox-2), Kruppel-like factor-4 (Klf-4), cellular Myc (c-Myc), and DNA-binding transcription factor NANOG in cancer stem cell models [87,88,89]. Cumulatively, these stem cell markers serve as specific and sensitive quantitative endpoints for characterizing stem cell populations and confirming the stem-cell-targeted efficacy of test agents.

Since the initial discovery of malignant stem cell populations within tumors, numerous studies have identified CSCs in various cancer types, including leukemia [90,91,92], breast cancer [93], colorectal cancer [94,95,96], prostate cancer [97], lung cancer [98], brain cancer [99], and melanoma [100]. Resistance of CSCs to traditional cancer treatments is associated with factors such as drug efflux proteins and proteins related to interleukin-4 (IL-4) signaling, as well as with the heightened activity of aldehyde dehydrogenase (ALDH) [101]. Furthermore, the atypical expression of genes within a variety of signaling pathways, such as the Janus-activated kinase/signal transducer and activator of transcription (JAK/STAT) pathway; the Hedgehog, Wnt, Notch, phosphatidylinositol 3-kinase/phosphatase and tensin homolog (PI3K/PTEN) pathway; and the nuclear factor-κB (NF-κB) pathway, has been observed in various CSCs, contributing to their resistance to drugs and treatments [95]. Phenotypically, CSCs exhibit unique characteristics, including self-renewal capabilities and slower cell cycle rates compared to other cancer cell populations [26]. Consequently, most anti-cancer drugs that target highly proliferative non-CSCs may allow quiescent CSCs to evade treatment, resulting in tumor recurrence and therapy failure (Figure 2). To address this challenge, extensive pharmacological efforts have been dedicated to identifying chemotherapeutic agents that selectively target CSC populations [29,102]. For instance, in a previous study utilizing a chemical screening approach, certain compounds, including etoposide, salinomycin, and abamectin, exhibited significant and selective toxicity toward breast CSCs [29]. One compound in particular, the potassium ionophore salinomycin, was found to reduce the proportion of CSCs by >100-fold relative to paclitaxel, a commonly used breast cancer chemotherapeutic drug, and inhibit mammary tumor growth in vivo [29]. Subsequent in vitro and in vivo studies further suggested that salinomycin has anti-CSC effects on other types of tumors, including osteosarcoma [103], melanoma [104], and prostate cancer [105]. Despite this compelling preclinical evidence, salinomycin has not received FDA approval for clinical cancer therapy due to concerns about its toxicity [106].

Notably, emerging evidence suggests that cancer cells within tumors have a dynamic ability to switch between a stem cell state and a differentiated state, which is referred to as cancer cell plasticity [107] (Figure 3). Related to this, two well-established processes, epithelial–mesenchymal transition (EMT) and mesenchymal–epithelial transition (MET), are recognized for their roles in the conversion of epithelial cells into mesenchymal cells and vice versa [108]. It is increasingly recognized that EMT and MET are not binary processes but rather dynamic and reversible transitions that can generate hybrid intermediate states with both epithelial and mesenchymal features [108]. These hybrid states can exhibit high plasticity and heterogeneity and can adapt to different environmental cues and therapeutic pressures. Therefore, EMT and MET are thought to be key mechanisms of cancer stem cell plasticity that enable tumor evolution and diversity [109]. In earlier studies, EMT has been identified as a key mechanism underlying cancer cell plasticity, where cancer cells transition from an epithelial morphology to a fibroblast-like mesenchymal state, gaining enhanced motility and invasive capabilities characteristic of stem cells [110,111]. This plasticity can be modulated by genetic variations, epigenetic modifications, or external cues that affect the tumor microenvironment, such as hypoxia, inflammation, or drug exposure [102]. Suggestive of the functional connection between the EMT process and the CSC phenotype, it has been observed that the tumor-initiating capacities of various cancer cell types, including breast cancer, melanoma, and glioblastoma cells, are heightened when transcription factors associated with EMT activation, such as the zinc finger E-box binding homeobox genes ZEB1 and ZEB2, are overexpressed [110,111,112,113]. In addition, the absence of E-cadherin, which serves as a guardian of the epithelial phenotype, is recognized as a pivotal hallmark of EMT [114]. The E-cadherin/β-catenin complex assumes a critical role in preserving the integrity of cell-to-cell connections among epithelial cells by regulating the Wnt/β-catenin signaling pathway [114]. Activation of Wnt signaling results in the disruption of the E-cadherin/β-catenin complex, releasing β-catenin and facilitating its translocation to the nucleus. Within the nucleus, it orchestrates the transcription of EMT-associated genes, including vimentin [115]. Besides tumor-intrinsic pathways, a growing number of studies suggest that growth factors, cytokines, and signals from cancer-associated fibroblasts (CAFs) and tumor-associated macrophages (TAMs) in the tumor-immune microenvironment, as well as hypoxia, can trigger EMT in CSCs [107,116,117]. Given the profound implications of EMT in cancer progression, anti-tumor agents targeting CSCs have been developed to obstruct or reverse the effects of EMT-related signaling and gene expression, ultimately inducing the dedifferentiation of CSCs [110,111].

On the other hand, growing evidence has suggested a connection between MET and stem cell-like characteristics, challenging the conventional perspective on the interplay between EMT and CSCs [109,118]. The prevailing notion suggests that the downregulation of EMT-related transcription factors (EMT-TFs) is essential to convert mesenchymal cells into epithelial cells, promoting increased proliferation and facilitating the formation of tumor metastases [113]. For malignant tumor cells to form clones, they must adopt an epithelial phenotype while maintaining a stemness state [119]. Notably, Padmanaban et al. uncovered that the rescue of E-cadherin expression, achieved by inhibiting TGFβ-receptor signaling, is crucial during the detachment, systemic dissemination, and seeding phases of metastasis in invasive breast ductal carcinomas [120]. Interestingly, Tsai et al.’s study distinctly supported the role of EMT in dissemination, with subsequent MET playing a key role in colonization and macrometastasis [121]. In addition, Ocaña et al.’s research also affirmed the involvement of EMT in dissemination and emphasized the necessity of reversing EMT for metastasis [122]. Although the exact role of each process in cancer evolution and metastasis is still being investigated, it is thought that both EMT and MET play important roles in the metastatic cascade, and that the balance between these two processes can influence the outcome of cancer [123].

## 4. Metabolic Plasticity of Cancer Stem Cells

Recently, extensive studies have established a strong link between metabolic reprogramming and cellular stemness, suggesting their pivotal role in CSC phenotype and plasticity as well as anti-cancer drug responses [21,25,124]. Glycolysis and mitochondrial oxidative phosphorylation (mtOXPHOS) are two primary metabolic pathways for generating cellular energy in the form of ATP [125,126,127]. Glycolysis occurs in the cytoplasm and is anaerobic (does not require oxygen), splitting one glucose molecule into two pyruvate molecules and yielding 2 ATP. In contrast, mtOXPHOS is an aerobic (requires oxygen) process that occurs in the mitochondria, in which pyruvate from glycolysis is further broken down to produce a substantial amount of ATP (30 to 32 ATP) through the tricarboxylic acid (TCA) cycle and the electron transport chain [128]. Unlike normal cells that primarily rely on mtOXPHOS for energy production, cancer cells prefer glycolysis over mtOXPHOS for energy production, even in the presence of oxygen, a phenomenon known as the Warburg effect, which was named after Otto Warburg [129,130]. This preference is attributed to the rapid energy production of glycolysis that meets the heightened metabolic demands of rapidly proliferating cancer cells, supplies essential metabolites for cell growth, adapts to low-oxygen tumor environments, contributes to immune evasion, and may lead to chemoresistance [130,131,132].

Notably, it is well established that pluripotent stem cells primarily rely on glycolysis for energy generation, in contrast to normal cells that predominantly use mtOXPHOS [133]. As exemplified by induced pluripotent stem cells (iPSCs), a transition from mtOXPHOS to glycolysis is observed as these cells attain stem cell pluripotency, indicating the integral role of metabolic shifts in the stem cell reprogramming process [134]. These findings suggest a close connection between metabolic reprogramming and stemness, with the glycolytic shift potentially playing a pivotal role in CSC development. The results of multiple studies support the notion that CSCs rely more on glycolysis than normal cancer cells. Similar to normal stem cells, glucose is a crucial nutrient for CSCs, and its presence within the microenvironment significantly augments the proportion of stem-like cancer cells within the cancer cell population. Glucose induces the expression of specific genes in CSCs that are related to glucose metabolism, such as GLUT-1, PDK-1, and HK-1/2, which contributes to CSC population expansion [135]. Accordingly, inhibiting glycolysis or depriving CSCs of glucose leads to smaller CSC populations. Compared to the majority of differentiated cells, small cell subsets with stem-like characteristics derived from various cancer cell lines, including glioblastoma [136], ovarian cancer [137], breast cancer [138], colon cancer [139], and osteosarcoma [140], have been found to rely more on glycolysis. As prototypical glycolytic cells, CSCs display significantly elevated glucose uptake, lactate production, glycolytic enzyme expression, and ATP levels when compared to non-CSCs [136,138,139]. In this regard, the stemness marker CD44 plays a pivotal role in regulating glycolytic metabolism [141]. Furthermore, glioblastoma CSCs, which heavily depend on glycolysis, demonstrate heightened migratory capabilities under hypoxic conditions [141]. Glycolysis has also been identified as the predominant metabolic state in radiotherapy-resistant stem cells within nasopharyngeal [142] and hepatocellular carcinomas [143]. Consequently, glycolytic metabolic reprogramming is a critical factor in CSC maintenance and is linked to malignant and therapy-resistant cancer evolution; thus, glycolytic pathways are considered a primary target for CSC-directed cancer therapy [21,25,144,145,146].

While the aforementioned studies suggest that CSCs predominantly rely on glycolysis, other research indicates that CSCs exhibit a preference for mtOXPHOS. A growing body of evidence has shown that quiescent or slow-cycling tumor-initiating CSCs exhibit lower glycolytic activity, reduced glucose consumption, decreased lactate production, and elevated ATP levels when compared to their differentiated cancer progeny cells in various tumor types, including blood cancer [147], glioblastoma [148], and pancreatic cancer [149,150]. Furthermore, breast CSCs exhibit elevated mitochondrial mass and membrane potential, leading to increased rates of oxygen consumption and chemo-resistance [151]. Notably, invasive cancer cells display heightened mitochondrial metabolism, driven by the expression of the transcriptional co-activator peroxisome proliferator-activated receptor gamma co-activator 1 alpha (PGC1α), which serves as the master regulator of mitochondrial biogenesis [152,153]. Accordingly, the inhibition of PGC1α diminishes the stemness properties of breast CSCs [154]. Moreover, NANOG, a pluripotency gene, drives tumorigenesis by directing metabolic reprogramming towards mtOXPHOS [155]. The heightened mtOXPHOS phenotype and elevated PGC1α expression appear to be associated with chemoresistance in CSCs [156,157,158]. Consequently, in contrast to normal stem cells and iPSCs, which primarily rely on glycolysis, CSCs display a divergent metabolic phenotype that can be either glycolytic or mtOXPHOS-dependent. Nevertheless, there is a mounting body of evidence that strongly indicates that, in both scenarios, proper functioning mitochondria are essential and pivotal for influencing CSC phenotypes, including stem-like properties, migratory capabilities, and resistance to pharmaceutical agents [21]. In this context, while the aerobic “Warburg” glycolytic phenotype has conventionally been deemed a distinguishing feature of malignant cancer cells, the existence of mixed findings indicates that tumor cells do not adhere to a single metabolic strategy to fulfill their energy requirements. Furthermore, it is noteworthy that the conflicting findings highlight the potential of CSCs to have remarkable metabolic adaptability, which enables them to switch between mtOXPHOS and glycolytic phenotypes in response to environmental cues and cellular signaling pathways. In line with this notion, there are reports indicating that CSCs can transition to the glycolytic metabolism when mtOXPHOS is inhibited [159,160], or switch to mtOXPHOS when glycolysis is suppressed [161]. Collectively, these results further emphasize the importance of adopting a drug treatment strategy that combines the inhibition of mtOXPHOS with therapy targeting glycolysis.

## 5. Anti-Cancer Mechanism of Vitamin C in Targeting Cancer Stem Cells

In recent years, in parallel with the increasing comprehension of the multiple mechanisms driving CSC heterogeneity and plasticity, a growing number of cancer studies at the subpopulation level have unraveled the potential epigenetic and metabolic mechanisms by which VC targets CSC evolution within various tumors [44,162,163] (Table 1). In the following sections, we will delve into recent noteworthy studies to provide more comprehensive insights into the anti-CSC/EMT effects of VC and the mechanisms that underlie VC’s action in both hematological and solid tumors.

### 5.1. Targeting Leukemic Stem Cells with Vitamin C

Previous normal stem cell studies have demonstrated that VC can maintain the proliferation of embryonic stem cells (ESCs) [164] and promote the reprogramming of somatic cells into iPSCs [165] by enhancing the activity of either Jumonji C (JmjC) domain-containing histone demethylases (JHDMs) [166] or TET DNA hydroxylases [167,168,169,170,171]. At the molecular level, VC was found to significantly enhance the production of 5 hydroxymethylcytosine (5 hmC) both in ESCs and during the reprogramming of mouse and human fibroblasts into iPSCs by activating TET DNA demethylase activity, which facilitates the conversion of 5-methylcytosine (5 mC) into 5 hmC [170,171]. Remarkably, TET proteins (TET1–3), particularly TET2, have been recognized as tumor suppressors in the hematopoietic lineage, with inactivating mutations occurring in a significant proportion of patients with myelodysplasia (MDS), acute myeloid leukemia (AML), and clonal hematopoiesis of indeterminate potential (CHIP), a premalignant condition found in approximately 10% of elderly individuals that increases their AML risk [172,173,174]. In line with this, genetic mouse model studies have demonstrated that TET1 deficiency leads to abnormal self-renewal and the expansion of hematopoietic stem cells (HSCs) with a B cell lineage preference [175], while TET2 deficiency results in a myeloid lineage bias [176,177,178]. Furthermore, combined TET1/TET2 loss restricts malignancy to the B cell lineage, while combined TET2/TET3 deficiency accelerates AML [175]. Notably, the TET protein deficiencies in these models lead to the loss of 5 hmC in HSC genomes, resulting in DNA hypomethylation that is linked to changes in lineage-specific gene expression and genomic instability associated with blood cancer development.

Suggestive of its crucial role as a co-factor for the anti-tumor activity of TET proteins, two recent studies have revealed that VC helps impede the evolution of blood stem cells, which is associated with the progression of leukemia [35,36]. Utilizing a metabolomic screening approach, one study discovered higher VC levels in human and mouse HSCs than in more specialized hematopoietic cell types, with the VC transporter SVCT2 being most abundantly expressed in HSCs compared to lineage-restricted progenitors and mature immune cells [36]. This study used *Gulo^−/−^* mice, which cannot synthesize their own ascorbic acid due to the absence of L-gulono-gamma-lactone oxidase (GULO), an enzyme that is critical for VC synthesis [179], to demonstrate that VC deficiency elevates HSC frequency and causes a loss of 5 hmC in the genome. These effects were reversible through dietary vitamin C intake, implicating deficient TET activity as the cause of the abnormal HSC expansion [36]. Furthermore, systemic VC deficiency (*Gulo^−/−^*) or the use of cell-intrinsic VC transporter knockout mice (*Slc23a2^−/−^*) was found to synergize with the Flt3ITD oncogene to accelerate leukemogenesis in bone marrow transplantation studies [36]. Correspondingly, VC deficiency exacerbated 5 hmC loss in HSCs with heterozygous or homozygous loss of *Tet2*, suggesting that a vitamin C-depleted micronutrient environment could globally impair the activity of TET proteins, including TET1 and/or TET3 [36].

In line with these findings, another study showed that the administration of VC closely mimicked TET2 restoration by amplifying the formation of 5 hmC in *Tet2*-deficient mouse hematopoietic stem and progenitor cells (HSPCs) [35]. VC treatment was also shown to restrain the formation of human leukemic colonies and the progression of primary human leukemia patient-derived xenografts (PDXs) [35]. In addition, VC was found to induce DNA hypomethylation and the expression of a TET2-dependent gene signature in human leukemia cell lines [35]. Given the emerging role of epigenetic dysregulation in driving malignancy, these findings underscore the potential of VC to inhibit the aberrant self-renewal of HSCs through its enhancement of TET DNA hydroxylase activity, thus highlighting its role as an epigenetic anti-cancer agent targeting leukemia stem cell evolution.

### 5.2. Targeting Liver Cancer Stem Cells with Vitamin C

Recent studies have unveiled the potential anti-CSC effects of VC, with a specific focus on hepatocellular carcinoma (HCC) and liver CSCs [37,38]. One study utilizing a combination of in vitro assays with cultured HCC cells and in vivo experiments involving HCC patient tumor samples showed that pharmacological VC (10 mM) induced cell death in liver cancer cells, with the response being closely linked to the expression of SVCT2. [37]. On a mechanistic level, the uptake of VC through SVCT2 led to an increase in intracellular ROS, subsequently causing DNA damage and ATP depletion, ultimately resulting in cell cycle arrest and apoptosis [37]. Interestingly, SVCT2 was found to be highly expressed in liver CSCs, and its expression was positively correlated with the expression of stemness-related genes, such as Sox-2, Oct-4, and the CSC marker CD133 [37]. The increased expression of SVCT2 enhanced the self-renewal properties of liver CSCs, rendering them more susceptible to pharmacological VC and resulting in significant reductions in tumor growth and the elimination of liver CSC populations in HCC cell line (Hepa1–6, HuH-7) xenografts and patient-derived xenograft (PDX) models [37]. Furthermore, a retrospective cohort study revealed a significant association between intravenous VC administration and enhanced disease-free survival (DFS) in HCC patients [37]. Aligning with the outcomes of this study, another recent investigation using liver cancer cell models (HuH-7, Hep3B) showed that pharmacological VC (1 mM) selectively suppressed the viability of both liver cancer cells and CSCs, resulting in decreased formation of cancer cell colonies and CSC-derived tumor spheroids as well as the inhibition of tumor growth in vivo [38]. Interestingly, pharmacological VC (4 g/kg) prevented liver cancer metastasis in a xenotransplantation model without suppressing stemness gene expression in liver CSCs [38]. Further experiments indicated that pharmacological VC elevated the concentration of H_2_O_2_ and induced apoptosis in these cells. These results suggest that the anti-liver cancer efficacy of pharmacological VC can be achieved through metabolic alterations, independent of stemness gene regulation [38].

### 5.3. Targeting Breast Cancer Stem Cells with Vitamin C

It has been increasingly recognized that CSCs possess a unique metabolic profile that sets them apart from non-CSCs and is essential for maintaining their stemness properties [180]. Reflecting the metabolic heterogeneity and adaptability of CSCs, a recent study employing the MCF7 breast cancer cell model has shed light on the ability of VC to selectively target CSC metabolism [39]. In experiments using multiple CSC probe systems for metabolic fractionation via flow cytometry, a subpopulation of MCF7 cells displayed heightened PGC1α activity, elevated mitochondrial ROS/H_2_O_2_ production, and increased NADH levels—distinctive features of the CSC metabolic phenotype that are indicative of higher mitochondrial biogenesis and metabolism [39]. Furthermore, in experiments using the mammosphere formation assay, a tool employed to assess the activity of putative breast CSCs in non-adherent in vitro cultures [181], these cells exhibited enhanced mammosphere formation capacity [39]. Intriguingly, VC was observed to induce oxidative stress and impede the activity of GAPDH, a pivotal glycolytic enzyme [39]. This inhibition not only affected the metabolic processes but also hampered mammosphere formation, with an IC-50 of 1 mM [39]. Based on this result, VC was found to be approximately ten times more potent than 2-DG, a classical glycolysis inhibitor, which has an IC-50 of around 10 mM when targeting CSC propagation [39].

On the other hand, a recent, similar study using triple-negative breast cancer cell models (MDA-MB-231 and MDA-MB-468) revealed differential sensitivities to high-dose VC (10~20 mM) based on differences in their cellular ROS scavenging capacities [40]. When MDA-MB-468 CSCs were exposed to a high-dose of VC, they exhibited higher resistance to ROS-induced damage, which was attributed to their elevated antioxidant activity, reduced mitochondrial damage, and smaller decrease in membrane potential (ΔΨm), when compared to MDA-MB-231 CSCs [40]. In addition, high-dose VC led to programmed cell death in MDA-MB-231 CSCs by activating the intrinsic apoptosis pathway, as indicated by the upregulation of cytochrome c, and caspases-9, -3, and -7, as well as PARP cleavage [40]. These results suggest that high-dose VC could serve as a potential strategy for targeting malignant breast CSCs, with their response being influenced by their individual internal antioxidant systems.

Another separate study involving the human breast cancer cell lines Bcap37 and MDA-MB-453 demonstrated that high-dose VC directly influences EMT pathways and the metastatic potential of cancer cells [182]. At concentrations of 0.01 and 0.1 mM, VC was found to promote cell migration and invasion in these cell lines compared to control cells, while at 2 mM VC, cell migration and invasion were notably suppressed [182]. The application of high-dose VC also led to the increased expression of the epithelial marker E-cadherin and the reduced expression of the mesenchymal marker vimentin, indicating the role of VC in inhibiting EMT in breast cancer cells [182]. Furthermore, high-dose VC effectively blocked TGF-β1-induced breast cancer cell invasion as well as reversed the TGF-β1-induced downregulation of E-cadherin and upregulation of vimentin in these cells [182]. Importantly, high-dose VC demonstrated a pronounced inhibitory effect on breast cancer metastasis in in vivo experiments [182]. Thus, these findings emphasize that VC could potentially serve as an anti-metastasis agent in breast cancer treatment.

### 5.4. Targeting Metabolic Plasticity in Pancreatic Cancer with Vitamin C

Previously, it was demonstrated that pharmacological VC selectively induces cytotoxicity and oxidative stress in multiple cancer cell types, including pancreatic cancer cells, while sparing normal cells [6,183]. Building upon this research, a recent study utilizing a pancreatic ductal adenocarcinoma (PDAC) model revealed that high-dose VC has a significant impact on the proliferation, viability, and metastatic potential of PDAC cells (8988T and 8902) through the inhibition of glucose metabolism and downstream regulation of EMT genes [75]. Intriguingly, both cell viability and colony formation assays demonstrated that VC at concentrations of 4 or 5 mM hindered pancreatic cancer growth while inducing apoptosis in a caspase-independent manner, as evidenced by the inability of zVAD-fmk, a well-known pan-caspase inhibitor, to prevent VC-induced apoptotic cell death. Further experiments have demonstrated that pharmacological VC inhibits glycolysis and the migration ability of PDAC cells by suppressing the Wnt/β-Catenin signaling pathway associated with EMT plasticity. In line with these findings, exposure to VC has been found to regulate the expression of EMT marker genes, particularly by downregulating the expression of the transcription factor Snail and its associated mesenchymal markers, consequently reducing PDAC metastasis. This is consistent with an earlier report indicating that exposure to high-dose VC suppresses the invasion and migration of breast cancer cells through the regulation of EMT marker expression, as mentioned above [182]. This evidence suggests that high-dose VC treatment can highly impact cell survival and metastasis via the metabolic reprogramming of EMT marker expression, offering a promising therapeutic target for future pancreatic cancer treatments.

### 5.5. Targeting Cancer Stem Cells with Vitamin C in Combination Therapy

The ability of tumor cells to adapt to conventional chemotherapy enables them to undergo phenotypic changes, leading to the acquisition of drug resistance, which, in turn, can result in treatment failure or tumor recurrence [184]. In this context, a recent study of various murine carcinoma cell models (CT26, MC38, 4T1) demonstrated that co-treatment with pharmacological VC (5 mM) potentiates the efficacy of anti-cancer nanodrugs and diminishes cancer cell stemness, thus preventing post-surgery recurrence and systemic metastasis [41]. Indeed, high-dose VC significantly potentiated the cytotoxicity of nanoscale coordination polymers (NCPs) delivering two clinical combinations of chemotherapeutics: carboplatin/docetaxel and oxaliplatin/SN38 [41]. In addition, co-administration of VC and NCP particles induced a metabolic shift in CSCs from glycolysis to mtOXPHOS that was accompanied by disturbances in mitochondrial dynamics and a decrease in the self-renewal potential of CSCs [41]. Such metabolic alterations induced by VC increased the sensitivity of CSCs to chemotherapy, thereby boosting the effectiveness of NCPs against resistant CSCs [41]. Furthermore, the combined treatment of VC and NCP particles effectively prevented the enrichment of CSCs induced by NCP treatment alone, as indicated by the significantly reduced expression of pluripotency factors (Sox2, Oct4, and Nanog) associated with cancer stemness [41]. As a consequence of CSC eradication, in subsequent in vivo xenograft experiments, the combined administration of VC (4 g/kg) and NCP particles not only prevented post-surgery recurrence in a colon cancer model but also effectively inhibited systemic metastasis in an orthotopic breast cancer model [41]. These preclinical findings suggest that pharmacological use of VC can not only enhance the therapeutic efficacy of chemotherapeutic nanomedicines against primary tumors but can also effectively address the significant limitations associated with conventional chemotherapy, such as drug resistance and tumor recurrence.

## 6. Conclusions

Beyond traditional bulk studies, recent advancements in cancer research have greatly expanded our comprehension of the intricate mechanisms underlying intratumoral heterogeneity (e.g., CSCs vs. non-CSCs) and plasticity (e.g., EMT) at the subpopulation level in multiple cancer types. This progress has been achieved by leveraging innovative approaches, such as 3D spheroid culture systems, single-cell analysis technologies, and the identification of lineage-specific CSC markers derived from normal stem cell research, under various physiological, pathophysiological, and drug-resistant conditions. Concurrently, a growing number of studies using VC have adopted similar methods, shedding light on how VC directly influences CSC/EMT characteristics by modulating epigenetic and metabolic pathways when administered individually or in combination with other anti-cancer agents. Overall, these studies indicate that VC, functioning as a redox-reactive molecule and/or a cofactor for DNA demethylating enzymes such as TET1/2/3, has a direct impact on the epigenetic and metabolic traits of CSCs as well as the expression of EMT-related genes. Furthermore, these findings emphasize the potential for VC to reprogram adaptive and drug-resistant plasticity within tumors, ultimately rendering them more susceptible to other anti-cancer treatments. Notably, recent research has increasingly drawn attention to the dynamic interplay between intrinsic tumor evolution and the tumor-immune microenvironment (TIME) [22,23,24]. In this regard, recent studies using animal models suggest that co-treatment of VC with dietary strategies, such as intermittent fasting (IF) [14], or cancer immunotherapy [15,16] can synergistically promote anti-tumor immune environments. Considering the multifaceted anti-tumor effects of VC as both a pro-oxidant and an antioxidant in the TIME, as recently reviewed in detail [17], it would be intriguing to investigate, using advanced single-cell or subcellular analyses (e.g., single-cell RNA/protein sequencing), whether VC used alone or in combination with anti-cancer regimens can impact CSC heterogeneity and plasticity by reconfiguring the intercellular and metabolic interactions of diverse immune and cancer cell populations within the TIME. Such comprehensive approaches are anticipated to provide novel insights and a solid rationale for incorporating VC into clinical cancer therapy, either as a standalone treatment or in combination with other anti-cancer strategies, such as dietary interventions, standard platinum-based chemotherapy, or cancer immunotherapy.

## Figures and Tables

**Figure 1 cancers-15-05657-f001:**
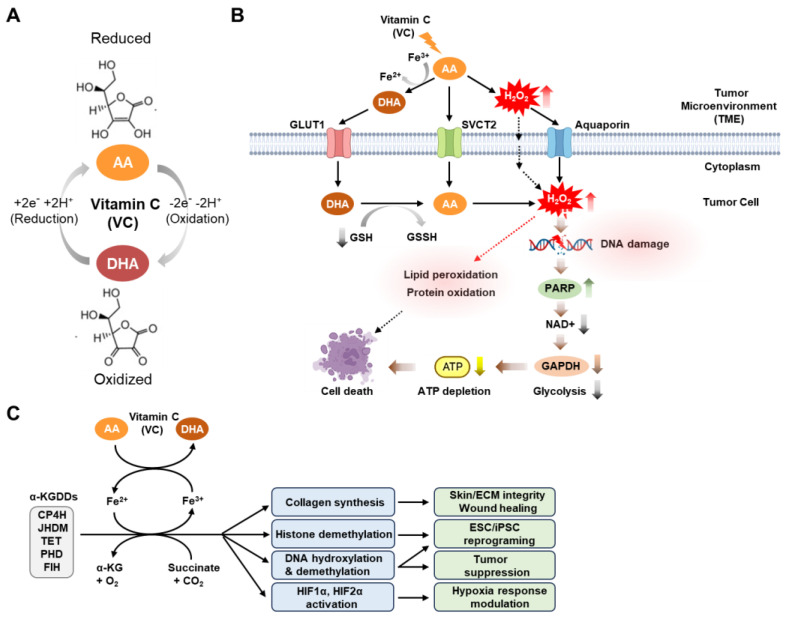
Physiological and anti-cancer mechanisms of vitamin C activity. (**A**) Physiological vitamin C (VC) exists largely in its reduced (ascorbic acid (AA)) or oxidized (dehydroascorbic acid (DHA)) forms, determined by either the gain or loss of two electrons and two hydrogens (reduction: +2e^−^ +2H^+^; oxidation: −2e^−^ −2H^+^). (**B**) Pharmacological VC can induce cancer cell death through two complementary mechanisms that elevate oxidative stress. Following VC treatment, hydrogen peroxide (H_2_O_2_) is produced in the extracellular environment by AA oxidation via Fenton chemistry that is facilitated by the presence of labile ferric iron (Fe^3+^) that enters cancer cells from the tumor microenvironment through either aquaporins or passive diffusion. VC enters cells through sodium-dependent vitamin C transporters (mainly SVCT2) when it is in its reduced form (AA), or via glucose transporters (mainly GLUT1) when it is in its oxidized form (DHA). Once inside the cell, dehydroascorbic acid (DHA) is rapidly converted to ascorbic acid (AA) through the action of the reducing agent glutathione (GSH). This process depletes the intracellular glutathione, resulting in elevated levels of intracellular H_2_O_2_ and several detrimental effects, including DNA damage, lipid peroxidation, and protein oxidation. In particular, DNA damage triggers the activation of the DNA repair enzyme poly (ADP-ribose) polymerase (PARP), which depletes cellular NAD^+^ levels. This depletion, in turn, inhibits the activity of glyceraldehyde 3-phosphate dehydrogenase (GAPDH) and glycolysis in cancer cells, resulting in decreased ATP production and cell death. (**C**) VC plays a pivotal role in numerous biological processes by serving as a cofactor for Fe^2+^ and alpha-ketoglutarate-dependent dioxygenases (Fe^2+^/α-KGDDs). These enzymes encompass a range of proteins, including collagen prolyl hydroxylases (CP4H), JmjC histone demethylases (JHDMs), ten–eleven translocation (TET) DNA hydroxylases, and hypoxia-inducible factor (HIF) hydroxylases (such as proline hydroxylase domain proteins (PHDs), and asparagine hydroxylase (factor-inhibiting HIF [FIH])). These enzymes have diverse functions, such as regulating collagen synthesis to maintain skin tissue and extracellular matrix (ECM) integrity as well as to facilitate efficient wound healing. They can also promote histone and DNA demethylation, thereby enhancing induced pluripotent stem cell (iPSC) reprogramming and suppressing leukemia progression. Furthermore, they can modulate various responses under low-oxygen conditions (hypoxia). This figure was created using BioRender, with modifications inspired by [42,43,44].

**Figure 2 cancers-15-05657-f002:**
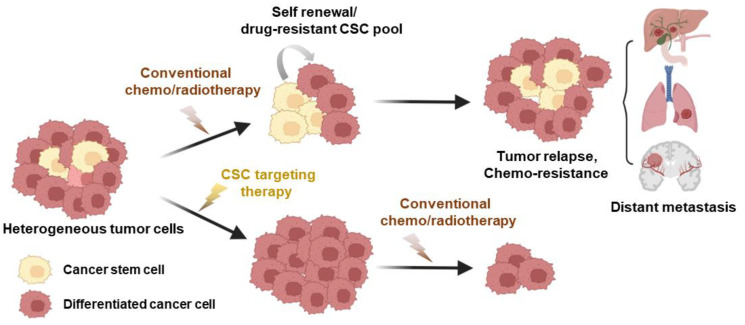
The influence of CSCs on anti-cancer treatment efficacy. Cancer stem cells (CSCs) represent a minority subpopulation within the overall tumor mass that displays remarkable resistance to chemotherapy and significantly contributes to tumor recurrence. Conventional treatments typically lead to a temporary decrease in tumor size by eliminating non-stem cancer cells (differentiated cancer cells). However, residual CSCs can give rise to recurrent tumors, and the initiation of metastasis is facilitated by the establishment of secondary cell colonies in distant organs. The adoption of CSC-specific inhibitors as cancer treatments has the potential to mitigate therapy resistance, lower the risk of relapse, and hinder metastasis, all while curtailing the stem cell properties of these cells. This figure was created using BioRender, with modifications inspired by [25].

**Figure 3 cancers-15-05657-f003:**
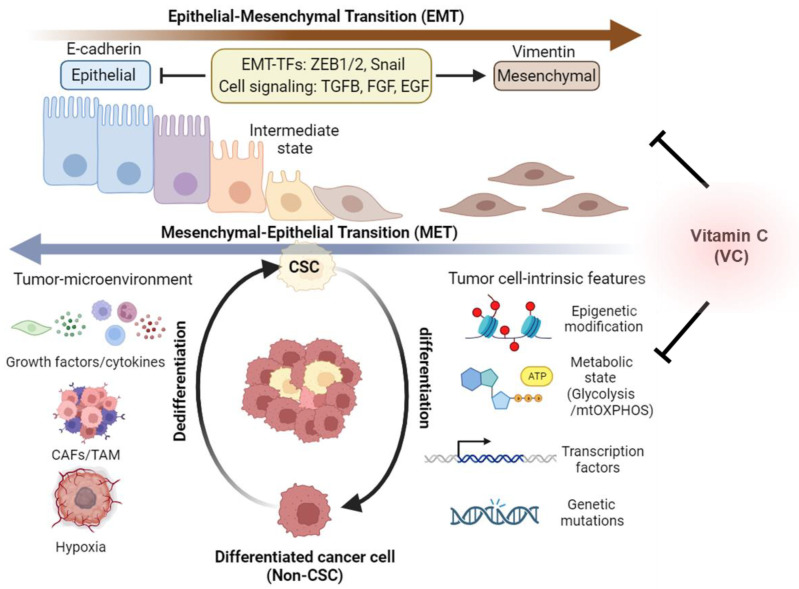
Impact of vitamin C on CSC heterogeneity and plasticity. Cancer cells exhibit intratumoral diversity via their ability to transition back and forth between CSC and non-CSC/differentiated states. Epithelial–mesenchymal transition (EMT) and mesenchymal–epithelial transition (MET) are two fundamental processes that mediate the reversible conversion of epithelial cells into mesenchymal cells and vice versa, which plays a crucial role in cancer cell plasticity and metastasis. These dynamic transitions are modulated by various factors, including epigenetic, metabolic, and genetic alterations within tumor cells, as well as by changes in the tumor-immune microenvironment. Additionally, CSCs are known to undergo an epithelial-to-mesenchymal transition (EMT), often adopting an intermediate EMT state. This transition is influenced by epigenetic modifications, metabolic reprogramming to shift from glycolysis to mitochondrial oxidative phosphorylation (mtOXPHOS), genetic mutations, and changes in the ways genes are activated or silenced in cancer cells. Moreover, signals emanating from the tumor microenvironment, such as growth factors, cytokines, the presence of cancer-associated fibroblasts (CAFs) or tumor-associated macrophages (TAMs), and hypoxia, also play a role in this transition. Vitamin C (VC) has been shown to regulate epigenetic and metabolic reprogramming and EMT marker gene expression, thereby impeding malignant CSC evolution. This figure was created using BioRender.

**Table 1 cancers-15-05657-t001:** The role of vitamin C in targeting cancer stem cell phenotypes and plasticity.

Cancer Stem Cell Type/Origin	Methods	Results	Ref.
Hematopoietic stem cells (HSCs) purified from the bone marrow of mice (Gulo^−/−^, Tet2^fl/fl^, Flt3ITD, Slc23a2^−/−^) or bone marrow aspirates collected from patients, aged 34–85, who were being assessed for lymphoma	Human hematopoietic cell purification Bone marrow reconstitution assays HSC culture Metabolomics to measure 5 hmC, 5 mC, and C by LC–MS/MS RNA-sequencing (RNA-seq) analysis	HSCs have high Vitamin C (VC) levels and ascorbate depletion increases HSC frequency. VC depletion reduces Tet2 activity in HSCs and progenitors in vivo. Low VC levels cooperate with Flt3ITD to promote myelopoiesis, in part, by reducing TET2 function, and cell-autonomously promote HSC function. Low VC levels accelerate leukemogenesis.	Agathocleous, et al. [36]
Primary mouse hematopoietic progenitor cells or bone marrow cells from TRE-TurboGFP-shTet2 and TRE-TurboGFP-shTet3 transgenic mice; Vav-tTA, Rosa-M2rtTA, and TRE-GFP-Ren mice; C57BL/6 B6.SJL-Ptprca Pepcb/BoyJ (CD45.1) mice; and germ-line Tet2-deficient mice Human leukemia cell lines: HL60, MOLM13, K562, KG1, THP1, and KASUMI1 Diagnostic bone marrow aspirates obtained from acute myeloid leukemia (AML) patients	Primary AML colony formation and liquid differentiation assays Bone marrow competitive transplantation Global DNA methylation quantitation, RNS sequencing, bisulfite sequencing analysis 5-hydroxymethylcytosine DNA immunoprecipitation (5 hmeDIP), sequencing, and analysis	TET2 restoration reverses aberrant self-renewal of Tet2-deficient cells. TET2 restoration promotes DNA demethylation, differentiation, and cell death. VC treatment mimics TET2 restoration to block leukemia progression. VC treatment enhances leukemia cell sensitivity to PARP inhibition.	Cimmino, et al. [35]
Human hepatocellular carcinoma (HCC) and mouse liver cancer cells Patient-derived xenograft (PDX) liver tumors from human patients	Colony formation assays with HCC cells (HCC-LM3 and HuH-7 cells) and liver CSCs Cell viability and cell invasion assays Knockdown of SVCT-2 via shSVCT-2 plasmid transfection SVCT-2 immunohistochemistry staining in HCC tumors Microarrays In vivo xenograft assays using the HCC PDX model and PDXs	SVCT-2 is highly expressed in liver CSCs and is required for the maintenance of liver CSCs. SVCT-2 determines the differential susceptibility to pharmacological VC-induced cell death. Pharmacological VC (10 mM) preferentially eradicates liver CSCs in vitro. SVCT-2-dependent mechanisms of pharmacological VC-induced cell death. Pharmacological VC (4 g/kg) impairs tumor growth and eradicates liver CSCs in vivo.	Lv, et al. [37]
Huh7 and Hep3B HCC cell lines	3D sphere formation and colony formation assays Cell viability analysis RT-qPCR H_2_O_2_ Assays In vivo xenograft assays	VC (0.5~1 mM) selectively inhibits the viability of liver cancer cells and liver CSCs in vitro. VC inhibits sphere formation and colony formation in liver cancer cells. VC (4 g/kg) prevents HCC xenograft tumor growth and metastasis in vivo.	Wan, et al. [38]
MCF7 human breast cancer cell line	CSC identification with a mitochondrial metabolism reporter (mPGC1α-eGFP-Puro-R) and NADH auto-fluorescence analysis 3D mammosphere formation assays Mitochondrial ROS/H_2_O_2_ detection assays Cell migration: in vitro scratch assays Metabolic flux analysis (MFA)	Mitochondrial biogenesis indicated by PGC1α reporter activity correlates with stemness. Increased mitochondrial ROS levels and H_2_O_2_ production contribute to stemness. Increased NAD(P)H levels directly correlate with stemness. VC (1~2 mM) blocks mammosphere formation.	Bonuccelli, et al. [39]
MDA-MB-231 and MDA-MB-468 Triple-negative breast cancer (TNBC) stem cells	Fluorescence-activated cell sorting (FACS) of CSC populations (CD44^+^/24^−^) Population doubling time (PDT)/cell proliferation assays Detection of ROS generation in CD44^+^/24^−^ CSCs via fluorescence microscopy and nitroblue tetrazolium (NBT) assays Mitotracker staining assays and JC-1 staining for qualitative assessments of mitochondrial integrity and membrane potential (ΔΨm)	Breast CSC yields are ~80% from TNBC cell lines with different morphologies and similar doubling times. Treatment with VC (10~20 mM) leads to changes in morphology followed by proliferation inhibition in breast CSCs. VC-induced ROS production and mitochondrial damage in sorted breast CSCs occurs in a dose dependent manner, with pronounced effects on MDA-MB-231 CSCs compared to MDA-MB-468 CSCs. The antioxidant activities/redox alterations that occur upon VC treatment are correlated with the VC sensitivities of the CSCs.	Sen, et al. [40]
CT26, MC38, and 4T1 murine carcinoma cells	Synthesis and characterization of nanocarrier particles (NCPs): Carboplatin (Carb)/Docetaxel (DTX) and Oxaliplatin (OX)/SN-38 (active metabolite of irinotecan) Flow cytometry analysis of pluripotency factors (SOX2, OCT4, and NANOG) 3D sphere formation assays Metabolic flux analysis with fluorescence lifetime imaging microscopy (FLIM)/GAPDH activity assays/mitochondrial morphology and membrane potential assessments In vitro apoptosis/cytotoxicity assays In vivo orthotopic xenografts and measurements of tumor growth/metastasis	VC (5 mM) enhances the cytotoxicity of NCPs against CSCs in vitro. VC transitions CSCs from glycolysis to mtOXPHOS and inhibits CSC self-renewal. VC (4 g/kg) potentiates the antitumor efficacy of NCPs and reduces tumor cell stemness in vivo. VC and NCPs in combination treatments prevent post-surgery relapse and inhibit systemic metastasis.	Jiang, et al. [41]

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
