# Peer review of "Role of Vitamin C in Targeting Cancer Stem Cells and Cellular Plasticity"

_cancers, 2023, doi:10.3390/cancers15235657_

Round 1
Reviewer 1 Report
Comments and Suggestions for Authors
I would like to thank the author for this excellent work. The topic of the manuscript is of high interest. The author comprehensively addressed the issue of targeting cancer stem cells and cellular plasticity. The work is well-written with language appropriate for academia and this type of article.
I have just one minor suggestion for the author:
I suggest adding short paragraph regarding drug-likeness of vitamin C (not only focusing on its mechanisms of action or the found efficacy as provided in the manuscript), address bioavailability (the bioavailability of synthetic and natural vitamin C has been subject of extensive research, wherein animal research have revealed disparities, while human studies have consistently demonstrated no statistically significant distinctions between the two variants. The bioavailability of vitamin C derived from fruits and vegetables, which serve as the main sources of this nutrient in the diet, is subject to the effect of several factors. These factors encompass the existence of additional nutrients and phytochemicals. Furthermore, what is stability of vitamin C? Can it be rapidly metabolized or excreted that may impair its use?) and toxicity (current anti-cancer therapies rely on the combination of agents – does vitamin C exert synergic and selective anticancer effects with conventional anticancer agents or lead to elevated toxicity that can limit their use? what forms of toxicity does vitamin C elicit?).
Are there any clinical trials (ongoing or planned) that investigate the effects of vitamin C as standalone treatment or in combined treatments?
Author Response
We are grateful for the positive and constructive feedback provided by reviewers. We have addressed concerns as follows:
Reviwer 1
I suggest adding short paragraph regarding drug-likeness of vitamin C (not only focusing on its mechanisms of action or the found efficacy as provided in the manuscript), address bioavailability (the bioavailability of synthetic and natural vitamin C has been subject of extensive research, wherein animal research have revealed disparities, while human studies have consistently demonstrated no statistically significant distinctions between the two variants. The bioavailability of vitamin C derived from fruits and vegetables, which serve as the main sources of this nutrient in the diet, is subject to the effect of several factors. These factors encompass the existence of additional nutrients and phytochemicals. Furthermore, what is stability of vitamin C? Can it be rapidly metabolized or excreted that may impair its use?) and toxicity (current anti-cancer therapies rely on the combination of agents – does vitamin C exert synergic and selective anticancer effects with conventional anticancer agents or lead to elevated toxicity that can limit their use? what forms of toxicity does vitamin C elicit?). Are there any clinical trials (ongoing or planned) that investigate the effects of vitamin C as standalone treatment or in combined treatments?
Thank you for your insightful comments. We appreciate your suggestion to include information on the drug-likeness, vitamin C bioavailability, toxicity issues, and clinical trials. Accordingly, we added the related paragraphs in the main text as follows;
“Notably, VC exhibits several drug-like properties that make it a promising therapeutic agent. VC has a low molecular weight (<500 Da, 176.12 g/mol), is water-soluble, and has a high oral bioavailability. The bioavailability of VC in foods is generally considered equivalent to the purified form within the recommended nutritional range of 15–200 mg [45]. Yet, this bioavailability diminishes by over 50% with higher amounts, such as doses exceeding 1000 mg. Since VC was first chemically synthesized in 1933, the bioavailability of synthetic and natural VC has been a subject of extensive research [46]. Animal studies indicate varying bioavailability between synthetic and natural VC, depending on the study design and animal model. In contrast, human studies have consistently shown no significant differences in their bioavailability [47]. While synthetic and natural VC share the same molecular and physicochemical properties, it has been reported that fruits and vegetables offer a wealth of micronutrients, dietary fiber, and phytochemicals that can modulate the absorption and utilization of VC [47,48]. Furthermore, VC is highly susceptible to oxidation and degradation, particularly in biological fluids such as plasma and blood [47,48]. The stability of VC in these fluids is influenced by several factors that can occur during processing (e.g., heat and light, pH, metal ions) and storage (e.g., temperature, oxygen exposure). Particularly, in vivo VC levels are determined by a balance between uptake, metabolism, and excretion (refer to [49] for a comprehensive review of these processes). Notably, the conversion of ascorbic acid to DHA in foods or the gastrointestinal tract can diminish the bioactivity of VC [49]. To enhance the chemical stability and bioavailability of VC, various chemically synthesized ascorbic analogs, such as ascorbate 2-sulfate, ascorbate 2-monophosphate, and ascorbate 2-triphosphate, have been developed [45]. Additionally, encapsulating VC in specific nanoparticles has been shown to improve stability during storage and delivery [45]." (Pages 3-4, lines 133-156).
“Aligned with its pleiotropic effects as a cancer-specific, pro-oxidative cytotoxic agent, anti-cancer epigenetic regulator, and immune modulator, high dose VC has been proposed as a potent adjuvant treatment for cancer, acting synergistically with numerous standard (chemo-) therapies and alleviating the toxic side effects of chemotherapy (refer to for recent comprehensive reviews [12,13] of this)." (Page 2, line 48-51).”
“Harnessing its multifaceted effects in cancer, high-dose VC has emerged as a promising therapeutic strategy, either as a standalone treatment or in combination with various standard (chemo-) therapies, potentially alleviating the toxic side effects associated with chemotherapy [12,13,83]. Notably, even intravenous administration of very high doses of VC, ranging from 1 to 200 g and administered repeatedly, was reported to be well tolerated in the majority of patients [84]. However, caution has been advised regarding the administration of high doses, as they may lead to overt side effects in certain susceptible patients, such as the formation of oxalate renal stones [85]. Moreover, it is important to note that some patients may experience side effects, including diarrhea, nausea, abdominal cramps, and other gastrointestinal issues [85]. In light of these considerations, National Clinical Trials (NCT) have been actively investigating the effects of vitamin C as a standalone treatment or in combined therapies across various cancers, including EGFR mutant non-small cell lung cancer (NSCLC) (NCT04033107), recurrent high-grade glioma (NCT01891747), metastatic colorectal cancer (NCT04516681, NCT02969681), KRAS and BRAF mutant colon cancer (NCT04035096), hepatocellular carcinoma, pan-creatic cancer, gastric cancer, colorectal cancer (NCT04033107), and acute myeloid leu-kemia (AML) (NCT02877277) (refer to a recent comprehensive review [17] for further details on this)” (Page 6, Lines 242-258).
Reviewer 2 Report
Comments and Suggestions for Authors
Comments to Author:
1. Introduction: It is notable that less toxic natural products including micro-nutrients, dietary phytochemicals and nutritional herbs commonly used in traditional Chinese medicine represent testable alternatives against therapy resistant cancers. This aspect is adequately addressed for vitamin C. Published evidence for preclinical efficacy phytochemicals and nutritional herbs and potential stem cell targeting effects should be included to provide a broader background for the readership. Recent reviews (Pharmaceuticals 14: 1318, 2021) may provide useful conceptual/technical background and relevant references. .
2. Section 3: This section on cancer stem cells should include published evidence on stem cell isolation and characterization and models developed from drug-resistant cancer stem cells. Recent review (Int. J. Mol. Sci. 23: 7055, 2022) may provide relevant references.
3. Epithelial-mesenchymal transition (EMT): It is notable that epithelial-mesenchymal transition (EMT) and mesenchymal-epithelial transition (MET) are mechanistically linked to metastatic evolution of chemo-resistant cancer stem cells. The section addressing EMT is well-written. This text needs to be revised to include conceptual/ technical background for MET More importantly, reciprocal expression status specific markers should be included.
Comments on the Quality of English LanguageMinor editing for English language is needed.
Author Response
We are grateful for the constructive feedback provided by reviewers. We have addressed concerns as follows:
Reviewer 2
- Introduction: It is notable that less toxic natural products including micro-nutrients, dietary phytochemicals and nutritional herbs commonly used in traditional Chinese medicine represent testable alternatives against therapy resistant cancers. This aspect is adequately addressed for vitamin C. Published evidence for preclinical efficacy phytochemicals and nutritional herbs and potential stem cell targeting effects should be included to provide a broader background for the readership. Recent reviews (Pharmaceuticals 14: 1318, 2021) may provide useful conceptual/technical background and relevant references.
Thank you for your constructive comments. Accordingly, we added the related paragraphs in the main text as follows.
“In this context, there is a growing recognition that less toxic natural products, possessing anti-cancer stem cell (CSC) activities, such as flavonoids, FDA-approved drugs derived from natural sources, and nutritional herbs commonly employed in traditional Chinese medicine, hold promise as potential alternatives for addressing therapy-resistant cancers [33,34]” (Page 2, Lines 81-85).
- Section 3: This section on cancer stem cells should include published evidence on stem cell isolation and characterization and models developed from drug-resistant cancer stem cells. Recent review (Int. J. Mol. Sci. 23: 7055, 2022) may provide relevant references.
Thank you for your helpful comments and information. We added the related paragraph in the main text as follows.
“Furthermore, diverse sophisticated assays have been devised for the isolation of CSC populations. These include fluorescence-assisted cell sorting of Rhodamine-123 positive side populations, detection of cell-membrane-specific antibody positivity, drug-efflux-based assays, identification of aldehyde dehydrogenase-1 (ALDH-1)-positive cells, and isolation of drug-resistant phenotypes [87]. Positive selection of resistant phenotypes in the presence of cytotoxic drug concentrations has been effectively utilized for isolating drug-resistant stem cells [34,88]. These cells exhibit characteristics such as stem-cell-selective tumor spheroid formation, cell surface molecules, and nuclear transcription factors. The status of stem cell markers is quantified through tumor spheroid formation and the expression of select molecules, including clusters of differentiation clusters of differentiation CD44, CD133 and nuclear transcription factors octamer-binding transcription factor-4 (OCT-4), sex determining region box Y-2 (Sox-2), Kruppel-like factor-4 (Klf-4), cellular Myc (c-Myc) and DNA-binding transcription factor NANOG in cancer stem cell models [87-89]. Cumulatively, these stem cell markers serve as specific and sensitive quantitative endpoints for characterizing stem cell populations and confirming the stem-cell-targeted efficacy of test agents.” (Page 6, Lines 267-283).
- Epithelial-mesenchymal transition (EMT): It is notable that epithelial-mesenchymal transition (EMT) and mesenchymal-epithelial transition (MET) are mechanistically linked to metastatic evolution of chemo-resistant cancer stem cells. The section addressing EMT is well-written. This text needs to be revised to include conceptual/ technical background for MET More importantly, reciprocal expression status specific markers should be included.
Thank you for the important comments. We appreciate your suggestion to include information on MET. We added the related paragraphs in the main text as follows;
“Two well-established processes, epithelial-mesenchymal transition (EMT) and mesenchymal-epithelial transition (MET), are recognized for their roles in the conversion of epithelial cells into mesenchymal cells and vice versa [108]. It is increasingly recognized that EMT and MET are not binary processes, but rather dynamic and reversible transitions that can generate hybrid intermediate states with both epithelial and mesenchymal features [108]. These hybrid states can exhibit high plasticity and heterogeneity and can adapt to different environmental cues and therapeutic pressures. Therefore, EMT and MET are thought to be key mechanisms of cancer stem cell plasticity that enable tumor evolution and diversity [109].” (Page 7-8, Lines 327-335).
“On the other hand, growing evidence have suggested a connection between MET and stem cell-like characteristics, challenging the conventional perspective on the interplay between EMT and CSCs [104,114]. The prevailing notion suggests that the downregulation of EMT-related transcription factors (EMT-TFs) is essential to convert mesenchymal cells into epithelial cells, promoting increased proliferation and facilitating the formation of tumor metastases [115]. For malignant tumor cells to form clones, they must adopt an epithelial phenotype while maintaining a stemness state [116]. Notably, Padmanaban et al. uncovered that the rescue of E-cadherin expression, achieved by inhibiting TGFβ-receptor signaling, is crucial during the detachment, systemic dissemination, and seeding phases of metastasis in invasive breast ductal carcinomas [117]. Interestingly, Tsai et al.'s study distinctly supported the role of EMT in dissemination, with subsequent MET playing a key role in colonization and macrometastasis [118]. In addition, Ocaña et al.'s research also affirmed the involvement of EMT in dissemination and emphasized the necessity of reversing EMT for metastasis [119]. Although the exact role of each process in cancer evolution and metastasis is still being investigated, it is thought that both EMT and MET play important roles in the metastatic cascade, and that the balance between these two processes can influence the outcome of cancer [120].” (Page 9, Lines 379-395).